

# Persistent mosquito fogging can be detrimental to non-target invertebrates in an urban tropical forest

Nicole S.M. Lee[1], Gopalasamy R. Clements[2], Adeline S.Y. Ting[1], Zhi H. Wong[3] and Sze H. Yek[1]

[1] School of Science, Monash University Malaysia, Bandar Sunway, Selangor, Malaysia
[2] Department of Biological Sciences and Jeffrey Sachs on Sustainable Development, Sunway University, Bandar Sunway, Selangor, Malaysia
[3] Malaysia Immersion Hub, Monash University Malaysia, Bandar Sunway, Selangor, Malaysia

## ABSTRACT

**Background:** Human population growth has led to biodiversity declines in tropical cities. While habitat loss and fragmentation have been the main drivers of urban biodiversity loss, man-made interventions to reduce health risks have also emerged as an unintentional threat. For instance, insecticide fogging to control mosquito populations has become the most common method of preventing the expansion of mosquito-borne diseases such as Dengue. However, the effectiveness of fogging in killing mosquitoes has been called into question. One concern is the unintended effect of insecticide fogging on non-target invertebrates that are crucial for the maintenance of urban ecosystems. Here, we investigate the impacts of fogging on: (1) target invertebrate taxon (Diptera, including mosquitoes); (2) non-target invertebrate taxa; and (3) the foraging behavior of an invertebrate pollinator taxon (Lepidoptera) within an urban tropical forest.

**Methods:** We carried out fogging with Pyrethroid insecticide (Detral 2.5 EC) at 10 different sites in a forest situated in the state of Selangor, Peninsular Malaysia. Across the sites, we counted the numbers of knocked-down invertebrates and identified them based on morphology to different taxa. We constructed Bayesian hierarchical Poisson regression models to investigate the effects of fogging on: (1) a target invertebrate taxon (Diptera) 3-h post-fogging; (2) selected non-target invertebrate taxa 3-h post-fogging; and (3) an invertebrate pollinator taxon (Lepidoptera) 24-h post-fogging.

**Results:** A total of 1,874 invertebrates from 19 invertebrate orders were knocked down by the fogging treatment across the 10 sites. Furthermore, 72.7% of the invertebrates counted 3-h post-fogging was considered dead. Our regression models showed that given the data and prior information, the probability that fogging had a negative effect on invertebrate taxa 3-h post-fogging was 100%, with reductions to 11% of the pre-fogging count of live individuals for the target invertebrate taxon (Diptera), and between 5% and 58% of the pre-fogging count of live individuals for non-target invertebrate taxa. For the invertebrate pollinator, the probability that fogging had a negative effect 24-h post-fogging was also 100%, with reductions to 53% of the pre-fogging count of live individuals.

**Discussion:** Our Bayesian models unequivocally demonstrate that fogging has detrimental effects on one pollinator order and non-target invertebrate orders,

Corresponding authors
Nicole S.M. Lee,
nlee0010@student.monash.edu,
nicole.leesm@gmail.com
Sze H. Yek, yek.szehuei@monash.edu

especially taxa that have comparatively lower levels of chitinisation. While fogging is effective in killing the target order (Diptera), no mosquitos were found dead in our experiment. In order to maintain urban biodiversity, we recommend that health authorities and the private sector move away from persistent insecticide fogging and to explore alternative measures to control adult mosquito populations.

# INTRODUCTION

Urban biodiversity is expected to decline under current human population growth rates. More than half of the world's population now resides in cities (*Zhang, 2016*)—this is likely to lead to massive land development and, consequently, greater rates of natural habitat loss and fragmentation (*Clark, Reed & Chew, 2007*). While urbanization has led to the decline of certain invertebrate taxa (*Eisenhauer, Bonn & Guerra, 2019*), it has resulted in an increase in incidences of vector-borne diseases such as Dengue fever and Malaria, especially in areas with poor planning and management practices (*Knudsen & Slooff, 1992*). Vector-borne diseases make up more than 17% of all infectious diseases and results in over one million deaths a year (*World Health Organization (WHO), 2017*). In particular, diseases spread by *Aedes* spp. mosquitoes such as Dengue, Chikungunya and Zika pose a serious health risk in cities due to the mosquito's affinity towards urban areas (*Koou et al., 2014*). Urbanization inevitably results in more breeding sites for these mosquitoes as stagnant water sources increase due to improper waste disposal practices, open trash cans, and poor surface-water drainage (*Lee et al., 2019*).

Malaysia is on the list of countries that have high incidences of Dengue outbreaks, with Dengue cases gradually increasing over the years (*European Centre for Disease Prevention & Control, 2019*). With limited vaccines available to minimize the spread of vector-borne diseases, prevention and control continue to be the main mitigation strategies (*Benelli, Jeffries & Walker, 2016*; *Fournet et al., 2018*). For mosquito-borne diseases, there are three main approaches: (1) chemical control that involves fogging (i.e., insecticide spraying) to kill adult mosquitos (*Usuga et al., 2019*); (2) biological control that uses natural predators of mosquito larvae; and (3) environmental management and integrated vector management to reduce the mosquito breeding grounds (*Amal et al., 2011*). Of these methods, fogging is the most common form of adult mosquito population control in Malaysia, and is mainly carried out by both the Ministry of Health and the private sector in urban areas that experience vector-borne disease outbreaks (*Amin et al., 2019*).

Studies examining the efficiency of fogging in controlling adult mosquito populations have yielded mixed results. Some demonstrate short-lived effective mosquito population control (*Amal et al., 2011*), but others show evidence of mosquito populations developing increasing resistance towards commonly used fogging insecticides (*Marcombe et al., 2011*; *Shafie, Tahir & Sabri, 2012*). The cost of mosquitoes developing

resistance to insecticides outweighs the benefits of temporary reductions in adult populations, especially when new reports of Dengue regularly emerge in recently treated areas (*Usuga et al., 2019*).

A major source of concern for urban biodiversity is that sanctioned insecticides used in fogging are not explicitly selective towards mosquitoes—this poses a serious threat to non-target invertebrate communities that share the same habitats as mosquitoes (*Braak et al., 2018*). For example, studies have shown that natural insecticides such as pyrethrins can kill a wide range of insects but are ineffective at killing its targeted species—mosquitoes (*Kwan et al., 2009*; *Abeyasuriya et al., 2017*). As such, more studies are needed to understand how fogging affects non-target invertebrates in the urban environment.

Here, we investigate the impact of mosquito fogging on: (1) its target invertebrate taxon (Diptera); (2) selected non-target invertebrate taxa; and (3) the foraging behavior of an invertebrate pollinator taxon (Lepidoptera) within an urban tropical forest.

## MATERIALS AND METHODS

### Study area

Our study was conducted in the Kota Damansara Community Forest (KDCF) (3.17°N, 101.58°E), a secondary forest located in Selangor, one of the most urbanized states in Malaysia (*Yaakob, Masron & Masami, 2012*). The forest is under the management of Forestry Department Malaysia (permit number for this experiment: PHD.ST.052/2019) and has a diverse invertebrate community. Over 13 different insect orders, mainly Coleoptera, Hymenoptera and Diptera were collected in a previous study (*Khadijah, Azizah & Meor, 2013*). As of September 2019, Selangor was the state with the highest reported cases of Dengue and Chikngunya disease in Malaysia (*European Centre for Disease Prevention & Control, 2019*). While the interior of KDCF is not fogged, its surrounding suburban areas are constantly fogged, making KDCF an ideal study site to examine the indirect effect of fogging on urban invertebrate (Fig. 1).

Tree's within KDCF primarily consist of secondary species such as *Alstonia scholarisa* and *Macaranga* spp., however, some primary species such as *Shorea platyclados* has also been observed in its premises (*Salleh, 2006*). Ten trees within the KDCF compound were chosen for fogging treatments (Table S1). The criteria for these trees are: (1) each tree is at least 100 m away from each other to prevent fogging overlap; (2) each tree is within the height range of three m for standardized vertical fogging dispersion; (3) each tree has an umbrella-like canopy cover with less than 10% herbivory damage on the canopy leaves for standardized horizontal surface area exposed to fogging; and (4) each tree is within one km away from the hiking trail as mosquitos tend so seek human hosts around hiking trails. Thus, it is likely that the chosen trees were subcanopy species.

### Fogging treatments

Insecticide fogging was carried out twice a week at 11.00 am across a total of 5 weeks in the months of August to September 2019. The fogging time for mosquito control should ideally be around dawn or dusk for most effective mosquito control (*Amal et al., 2011*).

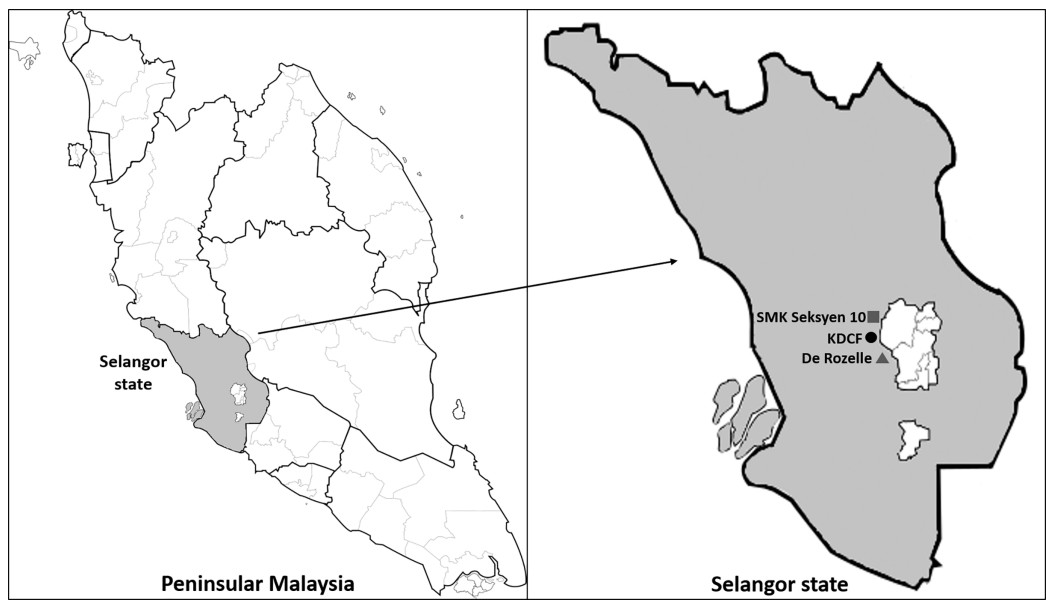

**Figure 1 The study site—Kota Damansara Community Forest Reserve (KDCF).** The entrance to KDCF (filled circle) is surrounded by a government school—SMK Seksyen 10 Kota Damansara (dark gray square), the high rise condominium—De Rozelle (dark gray triangle). KDCF experiences regular fogging by different private companies in an effort to control vector-borne mosquito diseases (Kota Damansara residents, 2019, personal communication). Image credit: OpenClipart at https://freesvg.org/.

However, for the purpose of our experiment, we choose 11.00 am as these are the times where most hikers are not using the KDCF trails and pollinators are most active. Nevertheless, we assumed that mosquitoes would be present in the canopy regardless of our fogging time based on a recent study conducted in KDCF (*Lee et al., 2019*). Professional fogging personnel from Ridpest Sdn Bhd (https://www.ridpest.com/) were employed to carry out the fogging experiment using a hand held pulse thermal fog generator (Fig. 2A). The Detral 2.5 EC insecticide brand, which consisted of the active ingredient deltamethrin 2.5% w/w, was utilized for the fogging treatment (Fig. 2B). Deltamethrin is a synthetic pyrethroid commonly used for mosquito fogging that targets the nervous system of invertebrates (*Chrustek et al., 2018*). This insecticide solution was prepared to the specified dosage (1:200 ratio of insecticide to water) according to instructions on the bottle label, used for normal fogging around residential areas. The licensed foggers would fog the tree starting at the bottom, thus allowing the fog to disperse to the top of the canopy (Fig. 2A). Each tree was fogged for 10 min, which is the minimum standard duration set by the Ministry of Health (http://www.moh.gov.my/). The standard duration for fogging is between 10 and 15 min depending on the severity of the mosquito-borne diseases reported and land area intended to be covered. For this experiment, the lower bound of the fogging time range as well as the insecticide used were chosen to simulate effects of conventional fogging practices for mosquitoes. KDCF itself is not normally fogged directly, but the study sites we chose were likely to experience spillover effects from nearby fogging and are thus ideal to investigate the indirect effects of

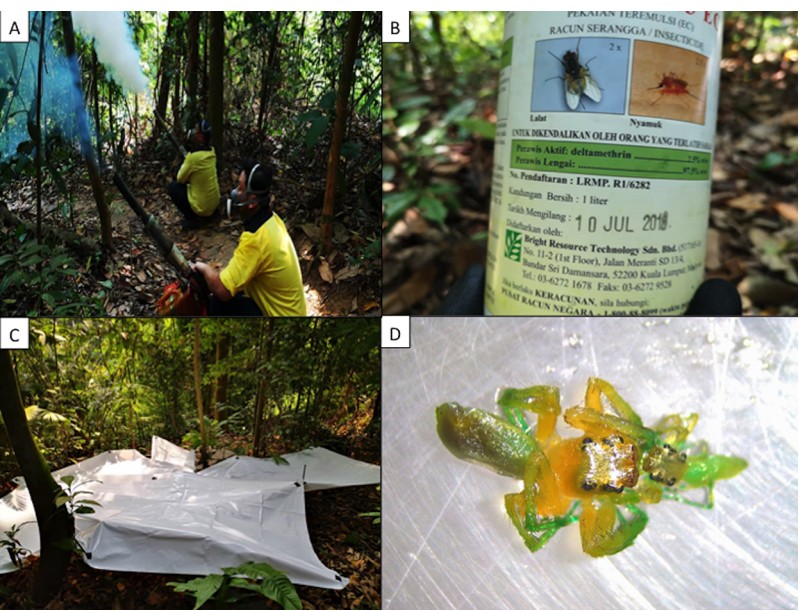

**Figure 2 Fogging experiments set-up and example of invertebrates.** (A) Licensed foggers using hand-held pulse thermal fog generators to fog one of the study site. (B) The fogging chemical Detral 2.5 EC brand used in this study. The active chemical (deltamethrin 2.5% w/w) is a form of synthetic Pyrethroid, claimed to be an effective insecticide targeting houseflies and mosquitoes. (C) Two 2.5 m and two 1.25 m polyethylene sheets set-up under the tree to fully cover the canopy of the site to maximize capture of knockdown invertebrate from the site. The sheets are held off the forest floor using 70 cm stakes to prevent leaf-litter invertebrates from crawling onto the sheets. (D) An example of dead invertebrate (order Araneae) due to fogging insecticide.

fogging (Fig. 1). Nevertheless, mosquito populations remain relatively high in the KDCF, as hikers often spray insecticide on their exposed body parts to ward off mosquito bites (E.L. Wong, 2019, personal communication).

To collect knocked down invertebrates, two 2.5 m and two 1.25 m white polyethylene sheets were set up under the tree corresponding approximately to the canopy cover (Fig. 2C). The sheets were held up off the forest floor by 70 cm stakes to prevent the leaf-litter invertebrates from crawling onto the polyethylene sheets. Invertebrates knocked down by the fog onto the sheets were carefully collected 5 min post fogging treatment into plastic containers that were covered with small nets for ventilation. Collected invertebrates were brought back to the lab for classification and sorting.

## Impact of fogging on target and non-target invertebrate taxa

To assess whether fogging was effective in killing its target invertebrate taxon (Diptera), and the extent to which it was detrimental to non-target invertebrate orders, we recorded their mortality 3-h after the fogging treatment. This time frame was chosen to understand the short-term effects of fogging on non-target invertebrate mortality rates. Invertebrates that responded to a light touch stimulus were categorized as "alive" and those that remained motionless as "dead". Each invertebrate was then sorted into their respective

orders based on their morphological characteristics with appropriate taxonomic keys (*McGavin, 1990*; *Imes, 1992*).

## Impact of fogging on foraging behavior of an invertebrate pollinator

To determine whether fogging was detrimental on the foraging behavior of invertebrate pollinators, we selected Lepidoptera as the focal taxon as they are important tropical pollinators, easily recognizable and also play a vital role as environmental indicators (*Tzortzakaki et al., 2019*). We counted the number of live butterflies occurring at each of the 10 sites pre- and post-fogging. On each day of the fogging treatment, the number of butterflies recorded in the 50 m radius of the site was recorded by two observers, each responsible for half of the radius. The counting of live butterflies was conducted for 30 min pre-fogging treatment. For post-fogging count, the same observation radius was repeated with the same observers, at the same site and time (approximately 10.00 am) for the same duration of time (30 min) 24 h after fogging treatment.

## Statistical analysis

We analyzed the data collected from 10 sample sites in a Bayesian framework to quantify the impact of fogging on counts of live individuals of: (1) Diptera 3-h post-fogging; (2) selected invertebrate taxa 3-h post-fogging (taxa selected based on detections in at least 8 of 10 sites); and (3) an invertebrate pollinator (Lepidoptera) 24-h post-fogging. We constructed Bayesian hierarchical Poisson regression models that are more suited for overdispersed count data (*Kim et al. 2013*). All analysis was conducted in R ver. 3.5.3 using packages "jagsUI" and "mcmcOutput".

## RESULTS

A total of 1,874 invertebrates were collected from 19 different orders after the 3-h post fogging treatments. An "Unknown" order consisting of 13 individuals could not be identified based on its morphological characteristics. These individuals are mostly immature forms of the invertebrates (Table 1). Of the total number of invertebrates collected, 72.7% (1,363) were knocked down by fogging and considered "dead", where Hymenoptera (18.0% of total knockdown insects) was the most abundant (majority were ants) and Diptera (8.8% of total knockdown insects) being the third most abundant order recorded as "dead" (Table 1). Out of all the Diptera individuals knocked down, no mosquitoes were collected, despite their presence verified by field researchers who were bitten by them during fogging experiments.

## Impact of fogging on target invertebrate taxon and non-target invertebrate taxa

Our regression models showed that given the data and prior information, the probability that fogging had a negative effect on invertebrate taxa 3-h post-fogging was 100%, with reductions to 11% of the original pre-fogging count of live individuals for the target invertebrate taxon (Fig. 3), and reductions to between 5% and 58% of the original pre-fogging count of live individuals for non-target invertebrate taxa (Fig. S1).

**Table 1 Summary statistics of knocked-down invertebrate taxa after the 3-h post fogging treatment across 10 sites in Kota Damansara Community Forest (KDCF), Selangor, Peninsular Malaysia.** The table is ordered from the most abundant to the least abundant knocked down invertebrate orders.

| Order | Number of knocked down invertebrates | Dead | Alive | Mortality 3 h post-fogging (%) |
|---|---|---|---|---|
| Hymenoptera | 337 | 217 | 120 | 64.4 |
| Araneae | 296 | 238 | 58 | 80.5 |
| Hemiptera | 209 | 144 | 65 | 68.9 |
| Thysanoptera | 208 | 159 | 49 | 76.4 |
| Coleoptera | 185 | 79 | 106 | 42.7 |
| Diptera | 166 | 148 | 18 | 89.2 |
| Collembola | 118 | 115 | 3 | 97.5 |
| Psocoptera | 112 | 106 | 6 | 94.6 |
| Acari | 63 | 47 | 16 | 74.6 |
| Blattodea | 51 | 38 | 13 | 74.5 |
| Orthoptera | 57 | 33 | 24 | 57.9 |
| Lepidoptera | 29 | 17 | 12 | 58.6 |
| Pseudoscorpiones | 10 | 2 | 8 | 20.0 |
| Archaeognatha | 5 | 4 | 1 | 80.0 |
| Neuroptera | 5 | 3 | 2 | 60.0 |
| Opiliones | 4 | 3 | 1 | 75.0 |
| Phasmotodea | 3 | 0 | 3 | 0.0 |
| Diplopoda | 2 | 1 | 1 | 50.0 |
| Mantodea | 1 | 0 | 1 | 0.0 |
| Unknown | 13 | 9 | 4 | 69.2 |
| Total | 1,874 | 1,363 | 511 | 72.7 |

## Impact of fogging on an invertebrate pollinator

Our regression models showed that given the data and prior information, the probability that fogging had a negative effect 24-h post-fogging was also 100%, with reductions to 53% of the original pre-fogging count of live individuals (Fig. 4).

## DISCUSSION

To our knowledge, this is the first study to demonstrate short-term detrimental effects of mosquito fogging on urban invertebrates in a tropical city in Southeast Asia. Our results demonstrate that the fogging insecticide had an unintended adverse effect on non-target invertebrates, which is characterized in this study as negative effects on invertebrates that were not mosquitoes. Similar results were also observed by *Abeyasuriya et al. (2017)* in Sri Lanka, where more dead than alive individuals were recorded amongst the 12 insect orders sampled 24-hr post fogging.

Our findings are, however, not concordant with previous studies that found that Diptera was among the most affected by fogging (*Kwan et al., 2009*; *Abeyasuriya et al., 2017*). In our results, Hymenoptera (consisting of ants, wasps and bees) was the most affected by

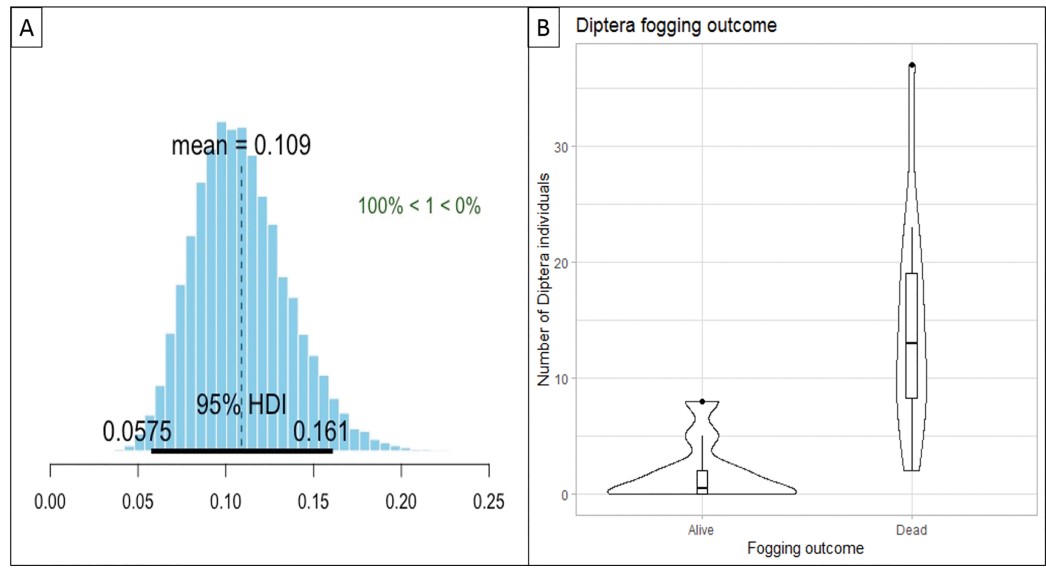

**Figure 3 Graphs displaying the impact of fogging on the target invertebrate taxon (Diptera) across 10 sample sites.** (A) Plot of marginal posterior probability distributions for our Bayesian hierarchical Poisson regression model, showing the probability that fogging had a negative effect on Diptera 3-h post-fogging was 100%, with reductions to 11% of the pre-fogging count of live individual, with the 95% Highest Density Interval (HDI) ranging between 5.8% and 16.1%. (B) A violin plot representing the distribution of "Dead" and "Alive" Diptera individuals found across the 10 sample sites. The distributions indicate that there are less "Alive" Diptera 3-h post-fogging. This can be seen in the larger distribution observed at the lower values of the "Alive" violin plot.   

fogging (Table 1). One possible explanation could be sites from both *Abeyasuriya et al. (2017)* and *Kwan et al. (2009)* studies had very different target and non-target invertebrate compositions, which are very dependent on the floral composition and the niches available at each site (*Toft et al., 2019*). At our study sites, the floral composition is of natural secondary forest composition, whereas *Abeyasuriya et al. (2017)* and *Kwan et al. (2009)* studies focussed on cultivated landscapes. Our results (Fig. 3) show that while the effectiveness of the insecticide in rendering more Diptera individuals "Dead" post-fogging is high, the selectivity of the insecticide towards mosquito species is low as none of the individuals were mosquitoes.

The unintended effect of fogging on non-target invertebrates is alarming as many of them play vital functions in urban ecosystems. Thysanoptera, for example, encompassing 11% of the total knocked down samples, was the sixth most affected order with 76.4% "dead" 3-h post-fogging. Commonly known as thrips, these invertebrates are important pollinators for many Dipterocarpaceae, an important hard-wood tree family that make up Southeast Asia's rainforest tree communities (*Apanah & Chan, 1981*). Thrips are also pollinators of *Macaranga* species (*Fiala et al., 2011*), an important pioneer tree genus for forest regeneration in Malaysia (*Daisuke et al., 2013*). An adverse effect on thrips diversity and numbers could severely disrupt pollination cycles of these two very important tree families, affecting existing dipterocarp tree biodiversity and might impede any forest restoration projects that plants *Macaranga* species.

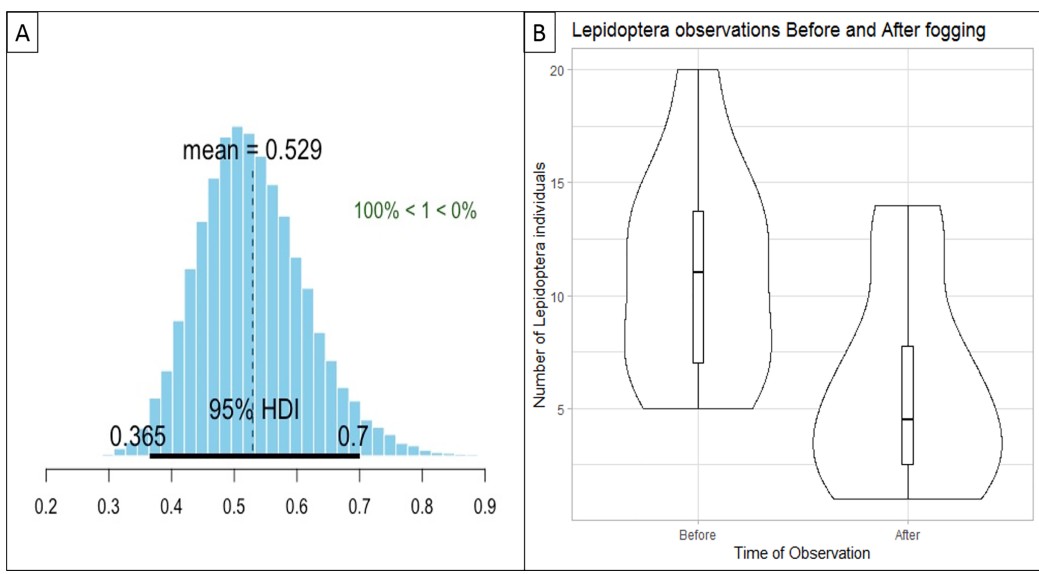

**Figure 4 Graphs displaying the impact of fogging on the invertebrate pollinator (Lepidoptera) across 10 sample sites.** (A) Plot of marginal posterior probability distributions for our Bayesian hierarchical Poisson regression model, showing the probability that fogging had a negative effect on Lepidoptera 24-h post-fogging was 100%, with reductions to 53% of the pre-fogging count of live individuals, with the 95% Highest Density Interval (HDI) ranging between 36.5% and 70%. (B) A violin plot representing the distribution of Lepidoptera observations "Before" and "After" fogging across the 10 sample sites. The distributions indicate that there are less Lepidoptera observations 24-h post-fogging treatment. This is observed where the distribution of data is larger at the lower values of the "After" violin plot.

Our study also reflects the varying degrees of insecticide susceptibility in invertebrates. Insecticide penetration may be less efficient in invertebrates with thicker cuticles and thus decrease their susceptibility to insecticides (*Dang et al., 2017*). Our results show that fogging appears to have more detrimental impacts on invertebrates with comparatively lower levels of chitinisation (e.g., live individuals of Psocoptera were reduced to 5% of pre-fogging count; Fig. S1). As recorded in other studies (*Boyce et al., 2007*), invertebrates with relatively "softer" bodies may permit easier entry of pyrethroids such as deltamehrin through contact as one of the primary modes of action (*Chrustek et al., 2018*). In contrast, Coleoptera, which have relatively higher levels of chitinisation due to unique adaptations of hardened forewings and compact bodies (*McGavin, 1990; Imes, 1992*), appeared to be more resistant to fogging (reduced to only 58% of pre-fogging count of live individuals; Fig. S1). Our results are consistent with a study by *Abeyasuriya et al. (2017)* where insects belonging to the order Coleoptera had the lowest mortality rate in two out of their three study sites. Even though hardened adult Coleoptera are more resistant to fogging insecticides, its larvae stages could still be affected.

Our findings indicate that fogging also has negative impacts on invertebrate pollinators such as butterflies. Sublethal exposure to insecticide may lead to changes in Lepidoptera foraging behavior and oviposition as the insecticides may alter the odor emitted by the plant (*De França et al., 2017*). This could be due to pollinators avoiding the insecticides that may be present in pollen and nectar (*Van der Sluijs et al., 2013*) or the fog has not
dispersed completely under the dense canopy. Furthermore, studies indicate that insecticides which target the nervous systems of invertebrates reduce pollinator survival and reproduction rates (*Abeyasuriya et al., 2017*; *De França et al., 2017*). While immediate fogging may not directly affect pollinators such as butterflies and bees, these organisms may become exposed to these chemicals through feeding and foraging (*Braak et al., 2018*) as pyrethroids have been shown to stick to pollen (*Pettis et al., 2013*). As evidenced from our study, most Lepidoptera individuals that were affected by fogging were caterpillars feeding on the vegetation when the fog hit. Future studies can focus on counting the number of Lepidoptera individuals in the fogged area for a longer period to investigate the extent they can recover to pre-fogging conditions. This result could give an indication of the length the fog persists on the surrounding vegetation. As our study only examined short-term effects of fogging on Lepidoptera, it is still unclear whether fogging has any long-term effects on pollinator behaviour or physiology.

In general, there is still a paucity of information on threats to invertebrate communities in urban areas. Most urban ecology studies have focused on the consequences of pollinator species decline (*Thogmartin et al., 2017*; *Meeus et al., 2018*; *Wepprich et al., 2019*), but very few studies have examined the consequences of general invertebrate decline. One possible consequence of the decline in non-target invertebrates is a negative effect on the survival of insectivorous birds, frogs, lizards and other invertebrate predators (spiders, wasps, etc.) that rely on invertebrates in their diet (*Sánchez-Bayo & Wyckhuys, 2019*). While fogging may not kill all invertebrates, the sub-lethal dosage exposed to these invertebrates may also have possible consequences on their biology, physiology and behavior (*De França et al., 2017*). Fogging may also lead to the homogenization of invertebrate species with generalist dominating the remnant habitat, reducing diversity and disrupting invaluable ecosystem services such as pollination, decomposition and nutrient cycling (*Sánchez-Bayo & Wyckhuys, 2019*).

## Caveats

Our results could have been more robust if we had adopted a Before-After-Control-Impact design, but limited resources were a constraint. We also acknowledge that our results were only reflective for the number of knocked-down insects that had dropped onto the collection sheets—they do not take into account the number of invertebrates, unaffected or affected, which remained in the canopy post-fogging. Future studies could account for this bias by sampling the canopy level and hidden crevices and leaves for a better representation of unaffected and affected invertebrates. Furthermore, as this study examines the short-term effects of fogging on non-target invertebrates, the cut-off timing for "Dead" or "Alive" categorization should be extended in future studies. This is to ensure that long-term effects can be captured by recording the number of invertebrates, initially recorded as "Alive" that eventually succumbed. *Abeyasuriya et al. (2017)* used a 24 h window as their cut-off point, and future studies could benefit by mirroring this 24-h period. While our study has documented invertebrates that are adversely affected by fogging, it would have been ideal to identify invertebrates to morphospecies to accurately determine differences in species diversity and richness affected by fogging. However, many

of the invertebrates collected were relatively small and of immature developmental stages where identification keys were absent. Metabarcoding could be explored in the future to obtain more accurate representation of species diversity. By doing so, the ecological functional groups of the invertebrates affected by fogging can also be identified.

## CONCLUSION

Overall, our study shows that insecticide fogging is detrimental to non-target invertebrates, particularly pollinators and species that have comparatively lower levels of chitinisation. Alternative methods of mosquito control should be explored in order to reduce health risks in tropical cities, while preserving other forms of urban biodiversity.

## ACKNOWLEDGEMENTS

We thank Ms. Harlina Binti Md Yunus and Forestry Department Malaysia for permission and logistics assistance. The fogging experiment could not run smoothly without the help from Ms. Deniece Yeo Yin Chia, Ms. Taneswarry Sethu Pathy and Mr. Jason Gan Yew Seng that assist in fogging experiments and subsequent invertebrate sorting. We also thank Ms. Wong Ee Lyn and Kota Damansara Community Forest (KDCF) regular hikers for inviting us to conduct this experiment at their managed forest to understand the effect of fogging on urban biodiversity. Finally, we are grateful to Mike Meredith for his advice on the statistical analyses.

### Funding

Support for this project by Sunway University (Project title: Direct and indirect effects of insecticide fogging on urban biodiversity; project code: INT-2019-SST-DBS-05) awarded to Gopalasamy Reuben Clements, Adeline Ting Su Yien, Wong Zhi Hoong and Yek Sze Huei. The funders had no role in study design, data collection and analysis, decision to publish, or preparation of the manuscript.

### Grant Disclosures

The following grant information was disclosed by the authors:
Sunway University.

### Competing Interests

The authors declare that they have no competing interests.

### Author Contributions

- Nicole S.M. Lee performed the experiments, prepared figures and/or tables, authored or reviewed drafts of the paper, and approved the final draft.
- Gopalasamy R. Clements conceived and designed the experiments, analyzed the data, prepared figures and/or tables, authored or reviewed drafts of the paper, and approved the final draft.

- Adeline S.Y. Ting conceived and designed the experiments, authored or reviewed drafts of the paper, and approved the final draft.
- Zhi H. Wong conceived and designed the experiments, performed the experiments, authored or reviewed drafts of the paper, and approved the final draft.
- Sze H. Yek conceived and designed the experiments, performed the experiments, prepared figures and/or tables, authored or reviewed drafts of the paper, and approved the final draft.

## Field Study Permissions

The following information was supplied relating to field study approvals (i.e., approving body and any reference numbers):

Field experiments were approved by the Forestry Department Malaysia (PHD.ST.052/2019).

## Data Availability

The raw data are available as Supplemental Files.

## Supplemental Information

Supplemental information for this article can be found online at http://dx.doi.org/10.7717/peerj.10033#supplemental-information.

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
