# Peer review of "Persistent mosquito fogging can be detrimental to non-target invertebrates in an urban tropical forest"

_PeerJ, doi:10.7717/peerj.10033_

## Round 0.1 · original submission · Major Revisions

Dear Dr. Lee and colleagues:

Thanks for submitting your manuscript to PeerJ. I have now received two independent reviews of your work, and as you will see, the reviewers raised a serious concern about the research. Despite this, these reviewers are optimistic about your work and the potential impact it will have on research studying the impact of mosquito fogging on forest arthropod diversity and population numbers. Thus, I encourage you to revise your manuscript, accordingly, taking into account all of the concerns raised by both reviewers.

Please consider the concerns about whether or not mosquitoes were present prior to fogging. This is a critical control and must be present in your revision. I see no way we can publish your work without having this control, unless the study is refocused on just the effects of fogging on arthropods in general.

Please address this major concern, as well as the other concerns of the two reviewers..

I look forward to seeing your revision, and thanks again for submitting your work to PeerJ.

Good luck with your revision,

-joe

Reviewer 1 ·

Basic reporting

The manuscript is generally well written. I note several areas where clarity of the manuscript could be improved. These are listed in order of importance.

1. Figures 3 & 4 do not value add substantially to the manuscript. It would probably be better presenting the information as a series of paired boxplots showing the (possibly log-transformed) abundances of alive vs dead for each invertebrate taxon considered. Statistical significance/credible-ness can simply be indicated using asterisks on such a plot. Nevertheless, see comment below regarding alive-vs-dead comparison

2. L25-31 of the Introduction seem a bit irrelevant to the topic of this study

3. The statistical analysis portion of the Materials and Methods (L158-168) was not clearly written. A "Bayesian framework" says nothing about the nature of the analysis used. Instead, the purpose and inferential mechanism of the analysis should be stated (e.g., this test is a quantitative comparison between two samples). The statement of L164 is also a questionable exaggeration and should be avoided (it is rather that BEST allows the relaxation of assumptions that classical t-tests require).

Experimental design

1. My greatest concern is that the alive versus dead comparison does not actually yield the desired quantitative insight into insecticide vulnerability/susceptibility
At least three quantities remain unknown here:
(1) the number of all invertebrates that were in the canopy (thus the “global” sample-able total)
(2) the number of all invertebrates that remained in the canopy—whether they died up there, were unaffected by fogging, or flew away when fogging commenced
(3) the number of invertebrates which were initially knocked down by insecticides, but which recovered and flew/crawled out of the canvas before the 3 hour mark
(4) the number of invertebrates that were recorded as “alive” which eventually succumbed.
(3) and (4) are not extremely important, although they should at least be addressed at some point in the manuscript (e.g., as a limitation). But it is (1) and/or (2) which have the potential to invalidate the conclusions of the study, because without a knowledge of these quantities, everything from L223-256 of the discussion section cannot be substantiated by the data—I am not saying these conclusions are false, but simply that they are not defensible by the data presented within this study. The alive versus dead at the 3-hr mark simply does not yield the desired quantitative insight into insecticide vulnerability/susceptibility that the authors seek to investigate.
It is however important to point out the flip side to this issue. In L219-222, authors say that “Our results demonstrate that the fogging insecticide had an unintended adverse effects on non-target invertebrates - this was also observed by Abeyasuriya et al. (2017) in Sri Lanka, where more dead than alive individuals were recorded amongst the 12 insect orders sampled post- 24-hr fogging.” But the “unintended adverse effects [of fogging] on non-target invertebrates” is not contingent on a alive-versus-dead ratio of >1. In an extreme case, if 10 butterflies were knocked down and 1 died while the remaining 9 recovered, non-target mortality has still occurred.

2. The study is meant to evaluate the effect of conventional fogging practices on non-target invertebrate taxa, presumably those that inhabit natural areas such as KDCF forest. However, authors seem to emphasize that the methodology for tree fogging is unconventional in its timing and location. It may be necessary to clarify how this methodology aligns with the study's objectives.

Validity of the findings

See point above regarding alive versus dead counts.

A key premise to the study, is that mosquitoes were present in trees when fogging occurred, but that fogging failed to knock these down. The conclusion suggested by authors was that mosquitoes may have developed insecticide resistance. I would encourage authors to strengthen this premise and perhaps even test this conclusion. For example, a simple manual search by hand (sweep netting) can be used to determine if mosquitoes are present in the canopies of trees in KDCF. The identity of these mosquitoes (Aedes or otherwise) would also lend additional evidence for or against the supposed conclusion.

Additional comments

The study is an ambitious one which addresses a topic that is very important both to larger society and the ecology and conservation of invertebrates in the natural environment. A significant methodological issue mars the study somewhat, but it nevertheless presents data that is important, and from which important insights can be drawn.

Reviewer 2 ·

Basic reporting

If this study’s goal is to replicate how Aedes aegypti are controlled in the study region regardless of being it is a sound practice, then I would consider it as an operational note and try to publish it with Society of Vector Ecology or American Mosquito Control Association journals as an operational note.

Experimental design

Please see comments in the general comments section.

Validity of the findings

One of the major problems interpreting the results is the lack of continued surveillance before and after thermal fogging applications in order to evaluate abundance for all species investigated in the MS. There isn't a section dedicated to explain how surveillance is conducted. So I have no clue about the surveillance data presented in the MS. What methods, how long, how often, duration? How did you sample Collembola, butterflies etc. before thermal fogging? Biggest challenge with non-target studies is that there are too many factors affecting population densities. It is difficult to tease out the affect of insecticides. For example, is there any evidence that non-target organisms investigated here declining when compared with historical abundance data?

Additional comments

Dear Authors,
I am afraid this study has a major problem with the experimental design, unless this is only an operational note which in that case will not fit the scope of PeerJ.
Although it has not been spelled out in the MS, I believe the study’s goal is to investigate thermal fogging to control dengue vector Aedes aegypti and while doing that what is the non-target effect? If you look at the limited studies on truck mounted-adulticiding and barrier treatments for Aedes aegypti and Aedes albopictus; they are geared towards treating low lying structure and shrubs. These locations are most often the host-seeking and resting ground for these species. For this reason, when trees are fogged you will not be able to observe dead mosquitoes, because live mosquitoes are not present there to begin with.
The concept of resistance is totally misinterpreted as adulticiding is an in efficient way to do mosquito control. We are aware of resistance and are constantly making progress with monitoring. There are numerous publications showing efficacious adulticiding applications. Also bear in mind, container-inhabiting mosquitoes are not the only target mosquito for mosquito abatement districts and professionals. Using adulticides work for salt marsh mosquitoes. Ae. albopictus is still more sensitive to pyrethroids and other adulticides and I can keep continue with more examples. Organophosphates are also effectively cause mortality for Ae. aegypti adults. However, the way this MS has been written it states adulticiding an ineffective control method for adult mosquitoes altogether.

---

## Round 0.2 · Minor Revisions

Dear Dr. Lee and colleagues:

Thanks for resubmitting your manuscript to PeerJ. I have now received one re-review of your work, and as you will see, this reviewer raised some concerns about the research. However, it is clear that your revision failed to address some original concerns of the other reviewer, who opted out of reviewing.

I do encourage you to use the criticisms raised by the reviewer to improve your work. I also implore you to entertain the concerns of Reviewer 2 as best as possible, making it clear what the focus of your study is and how it does have limitations in regard to its experimental design.

I look forward to seeing your revision, and thanks again for submitting your work to PeerJ.

Good luck with your revision,

-joe

Reviewer 1 ·

Basic reporting

Reporting is generally clear throughout.

Authors may consider briefly reporting some key numbers or statistics (not statistical tests, simply, numbers, as presented in Table 1). For example, stating that Hymenoptera was the taxon present in greatest numbers in canopies (and regarding Hymenoptera, it would be good if it could be clarified if the majority of these were ants, wasps or bees).

Experimental design

The authors did well in including a section on limitations in the revision. This section now highlights the weaknesses in experimental design that I had commented on in the first review.

Validity of the findings

The "therefore" statement of L286-289 is slightly overconfident in my opinion. It is possible, but again, the limitations highlighted by the authors significantly reduce the certainty of such a conclusion. for example, the authors deliberately chose a time of day when mosquitoes may be less likely to be successfully killed by fogging (L135-138). The conclusion is certainly possible, but it should be tailored to accommodate the uncertainty of the methodology of the study.

Was the paired nature of the observations in "Impact of fogging on foraging behaviour of an invertebrate pollinator" taken into account in the statistical analysis? If so, how it was done should be stated in the description of the analysis. Most readers (like me) would not be familiar with BEST and what it does, and would expect a paired T-test on log-transformed abundance.

Additional comments

The authors have done a good job in responding to my earlier comments.

Two things can still be done to improve the study:
1. More descriptive biological context can be provided
For example, the size and species of the trees used in this study can be provided. If trees are <3m tall, it is likely that they are subcanopy species, immature individuals of canopy species, or else trees that occur in light gaps or forest edges. This information can give readers a better perspective of the biological context and scale of the study, and allow comparison to other studies of canopy fogging

2. Support the inference that soft bodied taxa are more vulnerable with the data
This inference appears to be one of the key findings of the study, but it is actually not shown to be substantiated by the data. A simple statistical test of proportion mortality between soft-bodied and hard-bodied taxa can be done to support this.

---

## Round 0.3 · accepted · Accept

Dear Dr. Lee and colleagues:

Thanks for revising your manuscript based on the concerns raised by the reviewers. I now believe that your manuscript is suitable for publication. Congratulations! I look forward to seeing this work in print, and I anticipate it being an important resource for groups studying the impact of mosquito fogging on forest arthropod diversity and population numbers. Thanks again for choosing PeerJ to publish such important work.

Best,

-joe

Reviewer 1 ·

Basic reporting

The revision is significantly clearer than earlier versions of the manuscript

Experimental design

-

Validity of the findings

-

Additional comments

Authors have done well in addressing earlier concerns/comments.